# Monitoring of Long-Lasting Effects of Fumigation with Dimethyl Disulfide (DMDS) on Root-Gall Index, Root-Knots, Other Nematode Populations, and Crop Yield over Three Protected Cucumber Crops in Bulgaria

**Zhenya Ilieva [1,\*], Tanya Lazarova [1,2], Aleksander Mitev [3] and Arben Myrta [4]**

1 Institute of Soil Science, Agrotechnology and Plant Protection, 1331 Sofia, Bulgaria; tanyalazarova12@gmail.com
2 Faculty of Biology, Sofia University "St Kliment Ohridski", 1164 Sofia, Bulgaria
3 Sagea OOD, 4003 Plovdiv, Bulgaria; aleksander.mitev@sagea.com
4 Certis Europe B.V., 3500 AP Utrecht, The Netherlands; myrta@certiseurope.com
\* Correspondence: zhenya107a@gmail.com

**Abstract:** In order to evaluate the long-term effect of the new fumigant DMDS and its potential to be a part of IPM of root-knot nematodes, a two-year experiment was conducted on cucumber in a greenhouse in Bulgaria. In the first year, DMDS was applied in comparison with farmer nematicide strategy and untreated control in the first cucumber crop. After two consequent catch crops—lettuce in the winter period—a second cucumber crop followed in spring–summer. In this crop, the DMDS effect was studied with no treatment and was integrated into the post-planting strategy with *Trichoderma* spp. and garlic extract at plots previously treated with DMDS. The effect of DMDS was followed until 450 days after fumigation. Nematocidal efficacy of DMDS was 96% and the yield increased 65.5% in the first cucumber crop, and 80% and 100% in the second cucumber crop, respectively, when nothing was done or a combined strategy with *Trichoderma* spp. and garlic extract was followed. The DMDS effect on 44 non-target soil nematode genera was also followed. The negative impact of the fumigation was limited in time as non-target nematode communities regained previous crop levels in five months during our study. Therefore, DMDS can be properly included in soil IPM programmes and appears very promising for the protected crop industry in Europe.

**Keywords:** dimethyl disulfide; root-knot nematodes; *Meloidogyne arenaria*; fumigant; trophic groups; nematode food webs; greenhouse; integrated pest management

## 1. Introduction

In Bulgaria, the cultivation of vegetables provides high revenue to farmers. In 2014–2018, about 50,000 t of vegetables were produced mainly for the national market, over 87% of which was from protected cultivations [1]. Among the most important vegetables are cucumbers cultivated in greenhouses or plastic tunnels, which are followed by lettuce or green spices during the winter period. However, continuous cultivation of susceptible crops, along with relatively high and more constant soil temperatures under protected conditions, has favored an increase in root-knot nematode soil populations and, subsequently, the extent of damage they cause to host crops.

Root-knot nematodes (RKN) of the genus *Meloidogyne* are the most damaging plant-parasitic nematode affecting vegetables worldwide [2]. The economic impact of these nematodes is increasing in European Union (EU) because of the reduction in numbers and restrictions in use of chemical nematicides [3].

RKNs are widely distributed in protected cultivations in Bulgaria. Samaliev [4] reported that only eight of the 55 greenhouses were free from RKN in 17 regions of the country. Moreover, this author observed high to very high levels of root-knot infestation

at the end of vegetation season in over 45% of the observed protected facilities. More recently (2014–2015), we observed a high level of infestation in one-third of the 15 vegetable greenhouses in Blagoevgrad and Plovdiv regions (Ilieva, unpublished data).

In a comparative bibliographic review, Greco et al. [5] reported that the most effective control of nematodes in intermediate and highly infested protected crops was achieved with soil fumigant nematicides in comparison with all other chemical and non-chemical control means. Among the most effective fumigants was dimethyl disulfide (DMDS). The effect of DMDS on plant-parasitic nematodes has been investigated in several European countries, such as in France, Italy, and Spain, under protected and open field conditions for vegetables [6–9]. Currently, DMDS is in the pesticide registration process in EU by Arkema.

DMDS has limited persistence in the environment, no long-term toxicological effects and no adverse effect on the ozone layer. According to our knowledge, limited studies have been undertaken on the long-term effects of fumigation with DMDS on target and non-target organisms. Many nematode groups are well known as bio-indicators of soil health [10] and several analytical indices are widely used for assessing soil stress [11–13].

Therefore, the aim of this study was to evaluate the long-term effects of DMDS on crop yield, RKN populations and other soil nematode groups in cucumber cultivated in the greenhouse in comparison with the most common farmer control practices.

## 2. Materials and Methods

### 2.1. Site Description

A two-year (four successive crops: three key crops cucumbers and two winter lettuce) experiment was carried out in an unheated glasshouse in the vicinity of Sadovo town (South Central Region of Bulgaria—42°7′58″ N, 24°56′52″ E and 158 m a.s.l.). The greenhouse was established in 2007, had been cultivated with cucumber during the last seven years, and no heating was supplied in winter seasons for the last five years. The soil was silt loam with pH 7.7 and good drainage. According to the farmer, problems of RKN started to appear in 2012 and chemical control was conducted with oxamyl (Vydate EC). After 2016, only applications of fosthiazate (Nemathorin 10G) were used to control *Meloidogyne* spp.

### 2.2. Experimental Design

Two crops of cucumber were grown from April to October in 2018 and 2019, and two catch (winter) crops of lettuce in-between from late November 2018 until March 2019. The trial started with crop 1 (July–October 2018) according to a randomized complete block design with three replicates per treatment. The experiment was carried out at the same site following the scheme in Table 1.

**Table 1.** Experimental trial scheme 2018–2019.

| 1st Cucumber Crop (July–October 2018) | 2nd Cucumber Crop (April–June 2019) | 3rd Cucumber Crop (July–October 2019) |
|---|---|---|
| DMDS | No treatment | No treatment |
|  | *Trichoderma* spp. + garlic extract | No treatment |
| Fosthiazate | Oxamyl | No treatment |
| Untreated control | Untreated control | No treatment |

The cucumber plants were planted at the rate 18,000 plants/ha. Spacing between the plants in rows was 35 cm and between the rows at 120 cm. The catch crop was lettuce at the planting rate of 50,000 plants/ha. DMDS 94.1% EC was applied once at the beginning of the first experiment with "DOSATRON D30SPE.ARK" through a drip irrigation system. The fumigated plots were covered with DAF plastic film (Arkema Barrier Film) before fumigation and uncovered 18 days afterward, corresponding to seven days before transplanting. In the second cucumber cycle, two bio-products, *Trichoderma asperellum* (T25) 0.5% and *T. atroviride* (T11) 0.5% (Tusal WG), and garlic extract (NemGuard

99.9% SC) were applied in one-half of the experimental plots previously treated with DMDS EC before the first cucumber crop and the other half did not receive any treatment. In both cases, as market references were used, the nematicides were usually applied by the farmers: fosthiazate before the first cucumber transplanting (Nemathorin 10G) and oxamyl (Vydate 10EC) after transplanting the second cucumber crop.

More details of the experimental design are given in Table 2.

**Table 2.** Details of the three cucumber crops at the experimental scheme.

|  | **1st Crop** | **2nd Crop** | **3rd Crop** |
|---|---|---|---|
| Planting date | 30 June 2018 | 12 April 2019 | 10 June 2019 |
| Replicates | 6 | 6 | 6 |
| Plot size (m$^2$) | 168 | 168/84 * | 168 |
| Previous crop | cucumber | lettuce | cucumber |
| Cucumber cultivar | Pert | Delano | Delano |
| Nr. plants/ha | 18,000 | 18,000 | 18,000 |
| Last harvest date | 22 September 2018 | 14 May 2019 | 14 October 2019 |

Note: * the plots were in two in the second crop to evaluate DMDS alone or followed by the integrated post-planting strategy.

DMDS was applied at 400 L/Ha via drip irrigation only before the first cucumber cycle on 5 July 2018. *Trichoderma* spp. was used in the second cucumber crop at 0.5 kg/ha in three applications every two weeks (22 April, 6 May and 20 May 2019) while garlic extract was used at 4 Lit/ha in three applications every two weeks (29 April, 13 May and 27 May). Farm standards were: for the first crop Fosthiazate 40 kg/ha distributed manually as granules on 25 July 2018; for second crop was Oxamyl EC one application via drip irrigation on 22 April 2019. In the second cucumber crop, the plots were primarily treated with DMDS and divided into two equal parts, of which one half was treated with biological products and the other half was left untreated.

### 2.3. Root-Gall Index and Root Mapping

To assess the effects of the treatments on root infestation, the plants in each plot were uprooted at the end of each crop cycle. The roots were then carefully washed free of adhering soil and their gall indices were evaluated according to the 1–10 rating scale suggested by Bridge and Page [14]. Galling degrees were used to map nematode root damage levels in the field [15]. These maps were developed (i) on the previous cucumber crop on 27 June 2018; (ii) at the end of the first cucumber crop of our experiment on 17 October 2018; and (iii) at the end of the second cucumber crop of our experiment on 1 July 2019.

### 2.4. Yield

All cucumber fruits from each plot were harvested and weights and numbers of marketable fruits per plot were recorded for the total 10 pickings per cucumber crop, except the last.

### 2.5. Soil Sampling and Processing and Nematode Identification

Two bulk samples of 30 cores (d = 18 mm) in a zigzag manner, from the 0–20 cm soil profile per replicated plot, were taken on each sampling. In order to follow the effect of biocontrol treatment, bulk samples of 15 cores (d = 18 mm) in zigzag from 0–20 cm soil horizon per half replicate were taken on each sampling date from different variants of the trial in the second year (2019). First sampling: After removal of the previous cucumber plant residues and milling the soil (3–4 July 2018). Second sampling: 18 days after treatment (DAT) (23 July 2018). Third sampling: 104 DAT (17 October 2018, end of the first cucumber crop). Fourth sampling: 258 DAT (20 March 2019, end of the lettuce crop). Fifth sampling: 362 DAT (1–2 July 2019, before planting the second cucumber crop). Sixth sampling: 450 DAT (8 October 2019, last harvest of the second cucumber crop).

Each bulk sample was carefully mixed, weighed and two sub-samples of 100 cm$^3$ each were taken and weighed. Nematodes were extracted with centrifugation according to Coolen [16] in sucrose at the specific gravity of 1.8. Nematodes were counted in vivo under the inverted microscope IX Olympus [17] and fixed in FAA after gentle heating at 50 °C for 3 min. Part of the nematodes from each sample were mounted on permanent slides for identification. Separate sub-samples of 20 g soil were taken from each bulk sample and dried in an oven to estimate correction of dry soil content for each sub-sample. Nematode density was referred to in 100 g dry soil.

To identify the root-knot nematodes to species level, perineal patterns of the adult females were prepared and observed under a microscope and compared with those existing in the literature. Monoxenic laboratory populations of RKN were established on tomato (cv. Money Maker). DNA was extracted from 10 females separately using worm lysis buffer [50 mM KCl; 10 mM Tris pH 8.3; Gelatine 0.05%; Tween 20 0.45%; 2.5 mM MgCl and 60 μg mL proteinase K (Promega)]. Female specimens were squashed on slide with pipette tip in 15 μL WLB. Kneaded material was collected in a tube with 15 μL WLB and centrifuged at 10,410 RGF for 3 min and transferred to −80 °C for at least 15 min. DNA extraction was conducted by a hour incubation at 60 °C followed by 10 min at 95 °C. PCR amplification was performed according to Adam et al. [18] with two couple of primers: in 5S-18S rDNA region 194 (5′TTAACTTGCCAGATCGGACG 3′) and 195 (5′TCTAATGAGCCGTACGC 3′); and specific SCAR Far (5′TCGGCGATAGAGGTAAATGAC 3′) and Rar (5′TCGGCGATAGACA-CTACAAACT 3′). The 25 μL reaction (1 μL DNA extract and 12 μL REDTaq ReadyMix PCR Reaction Mix (Sigma-Aldrich, St. Louis, MO, USA) and 1 μL of each primer 10 μM) was performed in Auto Q Server Gradient Thermal Cycler (Quanta Biotech, Byfleet, UK). DNA fragments were separated by electrophoresis in a 2% TBE buffer agarose gel at 60 V.

Other soil nematodes were identified to genus level and separated in trophic groups [10] and functional guilds [12]. Some of early-stage juveniles of Rhabditidae were identified to family level but their position in trophic group and guild was clear—Ba1. To evaluate the effect of DMDS on soil nematode communities, the Shannon–Wiener diversity index (H′), Richness (G), Evenness (Eχ), Maturity (MI) and Plant-Parasitic Indices (PPI) [11,19] were calculated and food web analysis were also made [13].

### 2.6. Statistical Analysis

All data were subjected to analysis of variance (ANOVA) using the software Statistica 7.0 of StatSoft (TIBCO Software Inc., Palo Alto, CA, USA). If significant effects of the treatments were observed, the statistical significance of the differences between treatments was checked with Tukey's test ($p < 0.05$).

In case of data showing heterogeneous variances ($p < 0.05$) according to the Levene test of homogeneity of variances, data were prior transformed to Ln (X + 1) (recommended for numbers) or Arcsine square root % [Arcsin $\sqrt{}$ (%)] (recommended for percent values).

## 3. Results

### 3.1. Root-Knot Nematode Identification

The RKN population in the experimental site was identified as *M. arenaria* based on the morphology of perineal patterns and DNA fragment with 720 bp length obtained by amplification of 5S and18S rDNA region and 420 bp product as result of species-specific SCAR, respectively, of the specimens from monoxenic laboratory cultures.

### 3.2. Root-Gall Index and Root Mapping

RGI maps and the relative trial schemes at different stages of the study were presented in Appendix A, Figure A1a–f. The root-gall index on the previous cucumber crop, before starting our experiment, averaged 3.6 (2.3–5.2) on 1–10 scale and was rather uniform in different sectors of the greenhouse as indicated in the map (Appendix A, Figure A1a). Analysis of covariant of locality vs. date in different variants was insignificant.

At the end of the first cucumber crop (October 2018), the root-galling index (Table 3), root mapping (Appendix A, Figure A1c) and nematode juveniles in the soil (Figure 1) were negligible in the plots fumigated with DMDS, lower than the untreated control in those treated with fosthiazate and rather high in the untreated control plots. DMDS and fosthiazate treatments significantly reduced root infection with 96% and 52%, respectively, while untreated control showed a RGI of 5.0 (Table 3). At the end of the second cucumber crop (June 2019), the root-knot severity at harvest increased slightly compared to the previous crop, in the untreated control plots (6.5), remained low in the plots treated with oxamyl as per local farmer practice (1.9) and in the half-plots that did not receive any further treatment after fumigation with DMDS the year before (1.3), with no significant difference between them. In the half-plots previously fumigated with DMDS and continuing with the strategy of *Trichoderma* and garlic extract, cucumber roots remained totally free of nematodes (Table 4). This effect is visually better seen on the root-mapping done at the removal of the second cucumber crop (Appendix A, Figure A1e).

**Figure 1.** Dynamic of root-knot nematode second stage juveniles (J2) in the soil during the whole study period (DBA days before and DAA days after application of DMDS); Statistical significance of the differences between variants and treatments after Tukey's test ($p < 0.05$) was marked with different letters.

**Table 3.** Efficacy of the treatments in controlling the root-knot nematode (*M. arenaria*) and increasing yield of the first crop cucumber. Summer–autumn cycle (July–October 2018).

| Treatment (Nr. Application) | Root Damage | | Total Yield/Ha | |
|---|---|---|---|---|
| | Rgi (70 Dat) | % Reduction | M Tons | % Increase |
| Untreated control | 5.0 a | - | 19.4 c | - |
| Fosthiazate 10G 40 kg/ha (1) | 2.4 ab | 52 | 26.7 b | 37.6 |
| DMDS 94.1% 400 L/ha (1) | 0.2 b | 96 | 32.1 a | 65.5 |

Note: Averages of three replicates in the same column sharing a common letter are not significantly different in accordance with the Tukey's test at a 95% confidence level. Root damage was based on root-galling index (RGI) (0–10 scale) of cucumber plants at the last harvest. DAT: days after transplant.

**Table 4.** Efficacy of DMDS alone and its strategies with biopesticides in controlling the root-knot nematode (*M. arenaria*) in second crop cucumber in the same plots and greenhouse. Spring–summer cycle (April–July 2019).

| Treatments First Crop (Nr. Application) | Treatments Second Crop (Nr. Application) | Root Damage | | Total Yield/Ha | |
|---|---|---|---|---|---|
| | | RGI (80DAT) | % Reduction | M Tons | % Increase |
| Untreated control | Untreated control | 6.5 a | - | 10.3 b | - |
| Fosthiazate 10G 40 kg/ha (1) | Oxamyl 10 L/ha (1) | 1.9 b | 70.8 | 11.7 b | 13.6 |
| DMDS 94.1% 400 L/ha (1) | *Trichoderma* (3) + Garlic extract (3) 0.5 kg/ha + 4 L/ha | 0.0 c | 100 | 16.7 a | 62.1 |
| | No treatment done | 1.3 b | 80.0 | 15.5 a | 50.5 |

Note: Averages of three replicates in the same column sharing a common letter are not significantly different in accordance with the Tukey's test at a 95% confidence level. Root damage was based on root-galling index (RGI) (0–10 scale) of cucumber plants at the last harvest. DAT: days after transplant.

At the end of the third crop, DMDS applied only before the first crop and followed by three applications of garlic extract and *Trichoderma* spp. during the second crop, also significantly reduced the RGI in comparison with the farmer strategy and the untreated control (Table 5).

**Table 5.** Efficacy of DMDS alone and its strategies with biopesticides in controlling the root-knot nematode (*M. arenaria*) on the third crop of cucumber that did not receive any treatment before it. Summer–autumn cycle (July–October 2019).

| Treatments First Crop (Nr. Application) | Treatments Second Crop (Nr. Application) | Treatments Third Crop | Root Damage | |
|---|---|---|---|---|
| | | | RGI (96 DAT) | % Reduction |
| Untreated control | Untreated control | No treatment | 6.4 a | - |
| Fosthiazate 10 G 40 kg/ha (1) | Oxamyl 10 L/ha (1) | No treatment | 5.2 b | 18.8 |
| DMDS 94.1% 400 L/ha (1) | Trichoderma (3) + Garlic extract (3) 0.5 kg/ha + 4 L/ha | No treatment | 3.7 c | 42.2 |
| | No treatment done | No treatment | 5.1 b | 20.3 |

Note: Averages of three replicates in the same column sharing a common letter are not significantly different in accordance with Tukey's test at a 95% confidence level. Root damage was based on root-galling index (RGI) (0–10 scale) of cucumber plants at the last harvest. DAT: days after transplant.

### 3.3. Yield

The nematocidal effect of the treatments greatly affected the yield of both the first and second cucumber crops. On the first crop (Table 3), yield increase versus untreated was 37.6% in plots treated with fosthiazate and rose to 65.5% in those treated with DMDS. On the second crop (Table 4), the yield of cucumber did not increase significantly from untreated in the plots treated with oxamyl (following the first crop application with fosthiazate), but continued to increase significantly in the half-plots fumigated before starting the previous crop with DMDS (50.5%) but no further treatment before the second crop and in the half-plots treated previously with DMDS and thereafter with *Trichoderma* spp. and garlic extract before the second crop (62.1%). No yield evaluation was recorded during the third crop.

### 3.4. Root-Knot Nematode Invasive Stages in Soil

The nematode extraction from soil samples collected before starting the trial and counting of second stage infective juveniles of *M. arenaria* indicated an even distribution of the nematode across the greenhouse (Figure 1), confirming the validity of the root-mapping method and making the trial site useful for the experiment.

At the end of the first cucumber crop (October 2018), the nematode juveniles in the soil were negligible in the plots fumigated with DMDS, lower than in untreated control in

those treated with fosthiazate and rather high in the untreated control plots (Figure 1), in agreement with the RGI and root mapping evaluations done at the same time.

At the end of second cucumber crop (June 2019), the juveniles in the soil showed very limited numbers since 18 DAT in the soil of the plots treated with DMDS and thereafter also in those treated with *Trichoderma* spp. and garlic extract with no significant differences between the two differently treated halves, again in agreement with the visual observations of RGI and root mapping at the same time.

### 3.5. Monitoring Other Nematode Populations

Information on the nematode community and impact of human activities on it is important to understand the effects of farmer practices, including control means. In the studied greenhouse four other plant-parasitic nematodes (*Paratylenchus* spp., *Criconemella* spp., *Tylenchorhynchus* spp., *Psilenchus* spp.) were isolated in low level (Table A1 in Appendix B) and DMDS had significantly negative impact on them until the last sampling—450 DAA.

The effect of DMDS on the soil nematode community was investigated calculating several indices (Table 6). Differences between variants were outlined with taxonomic richness (G) and connected with Simpsons' diversity index that revealed a clear reduction of the nematodes 18 days after application of DMDS. The diversity of nematodes was relatively low as only 44 genera were identified. Most of them belonged to the group of bacterial feeders (18 genera), followed by omnivorous (8), animal predators (7), fungal feeders (6) and plant-parasitic nematodes (5) (Tables A1–A5 in Appendix B).

**Table 6.** Diversity and functional indices of nematode community in different trial treatments during the study period.

| Treatment | 1 DBA | 18 DAA | 104 DAA | 252 DAA | 362 DAA | 450 DAA |
|---|---|---|---|---|---|---|
| **Taxonomical richness** | | | | | | |
| Untreated | 31 a | 37 a | 9 bc | 16 ac | 25 a | 31 a |
| Non-fumigant nematicide | 28 a | 37 a | 10 ac | 21 a | 26 a | 29 a |
| DMDS | 30 a | 5 b | 11 ac | 23 a | 26 a | 28 a |
| **H'—Shannon–Wiener diversity index** | | | | | | |
| Untreated | 2.0 a | 2.3 a | 1.2 b | 1.6 a | 1.9 a | 2.1 a |
| Non-fumigant nematicide | 1.8 a | 2.4 a | 1.0 b | 1.3 ab | 2.0 a | 1.7 a |
| DMDS | 1.9 a | 1.0 b | 1.2 b | 1.8 a | 1.9 a | 1.9 a |
| **E$\chi$—Evenness** | | | | | | |
| Untreated | 0.6 a | 0.6 a | 0.5 a | 0.6 a | 0.6 a | 0.6 a |
| Non-fumigant nematicide | 0.5 a | 0.7 a | 0.4 a | 0.4 a | 0.6 a | 0.5 a |
| DMDS | 0.6 a | 0.6 a | 0.5 a | 0.6 a | 0.6 a | 0.6 a |
| **MI—Maturity index** | | | | | | |
| Untreated | 1.8 ac | 1.6 a | 1.2 b | 1.8 ac | 1.5 a | 1.7 ac |
| Non-fumigant nematicide | 1.8 ac | 1.6 a | 1.1 b | 2.0 c | 1.5 a | 1.8 ac |
| DMDS | 1.9 c | 1.9 ac | 1.6 a | 1.6 a | 1.6 a | 1.9 ac |
| **PPI—Plant-parasitic index** | | | | | | |
| Untreated | 3.0 a | 2.9 a | 3.0 a | 3.0 a | 3.0 a | 3.0 a |
| Non-fumigant nematicide | 3.0 a | 2.9 a | 3.0 a | 3.0 a | 3.0 a | 3.0 a |
| DMDS | 2.9 a | 3.0 a | 3.0 a | 3.0 a | 2.9 a | 2.9 a |

Note: DBA: days before DMDS application; DAA: days after DMDS application. Non-fumigant nematicide: fosthiazate at first crop and oxamyl at second crop. Within each index statistical significance of the differences between variants and treatment after Tukey's test ($p < 0.05$) was marked with different letters.

The general abundance of free-living nematode groups was more even before treatment than with *Meloidogynes* (Figure 2). On the 18th DAA, clear difference between DMDS and other plots appeared and only a few bacterial feeding nematodes hatched in the DMDS variant. On the 104 DAA, the general density of "other" nematodes was still low in DMDS plot in comparison with control plots. The most abundant and frequently occurring nematodes genera were the bacterial feeders *Zeldia*, *Diploscapter* and *Cruznema*. These genera and *Meloidogyne* were the only ones that were isolated from DMDS-treated plots 18 DAA (Table A2 in Appendix B).

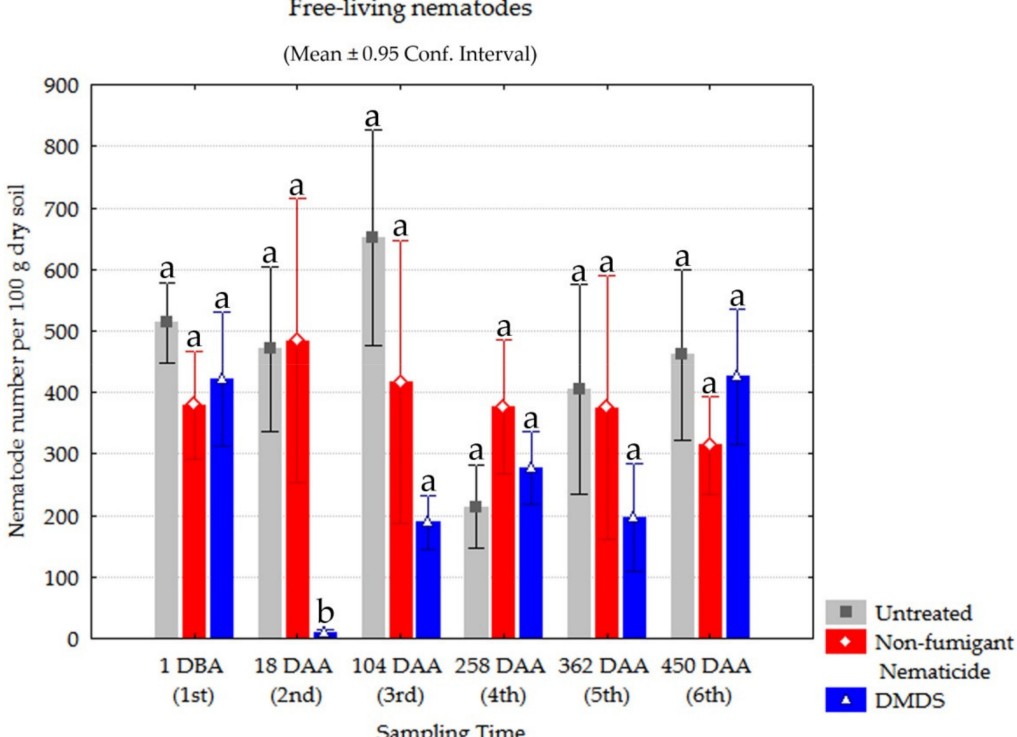

**Figure 2.** Dynamic of free-living nematode in the soil during the whole study period (DBA days before and DAA days after application of DMDS). Statistical significance of the differences between variants and treatments after Tukey's test (*p* < 0.05) was marked with different letters.

Fungal feeders were with low density in DMDS fumigated plots from the first sampling until the end of the study period. They were a little more abundant and fluctuated insignificantly over the five crops (three cucumber crops and two lettuce crops in the winter period 2018–2019) in the other treatments. All genera found in the greenhouse belonged only to Fu2. This data indicated that the main decomposition went through "fast" channels dominated by bacteria. Omnivorous nematodes had low abundance in all treatments and were absent after DMDS fumigation but their diversity and abundance recovered 258 days later (4th sampling). Animal predators were rare and with limited abundance in the studied area during the whole study period.

The use of the maturity index as a measure of stresses is not very useful in our study (Table 6). It did not significantly differ between the variants after DMDS application but showed differences in other samplings that had no explanation. Taxonomical richness and the Shannon–Weiner diversity indexes demonstrated early stress conditions at 18 DAA. Evenness and PPI did not indicate significant differences. Generally, the evenness was low in the studied greenhouse.

An evaluation of soil food webs based on weighed nematode faunal analysis is presented in Figure 3. Generally, food webs in the experimental area were characterized as disturbed, with low C:N ratio and dominated by bacterial decomposition channels (Figure 3). All samplings in untreated plots were situated in quadrant A (Figure 3a). Clearly, the position of data in DMDS treated plots 18 DDA in quadrant D corresponded with stress, depleted enrichment and degradation of food web condition (Figure 3c). In these plots, in all following samples, the food web was similar to those in untreated plots. In the plot treated with the nematicide used by a farmer, a winter degradation of the food web was observed on the 258th DDA of DMDS (winter samplings after two catch crops) (Figure 3b). In these experimental plots, high variability was observed and even on the last sampling, there were points in quadrant D.

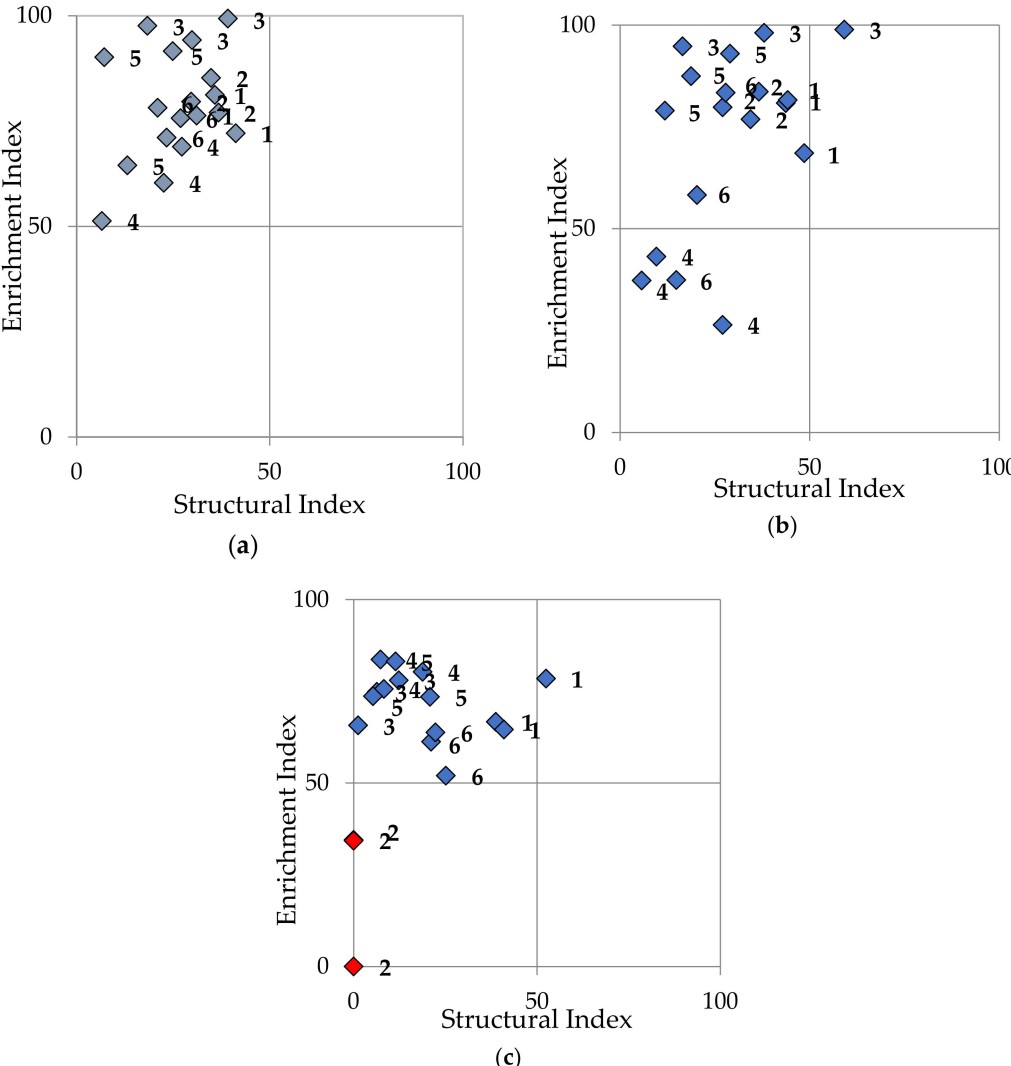

**Figure 3.** Soil trophic webs in variants ((**a**)—Untreated; (**b**)—Treated with conventional nematicide; (**c**)—Treated with DMDS) during whole study period (Sampling: 1—1 DBA; 2—18 DAA; 3—104 DAA at the end of Crop 1; 4—258 DAA after winter period, at the end of crop 3; —362 DAA at the end of crop 4, 6—450 DAA, at the end of crop 5).

## 4. Discussion

The efficacy of DMDS on the control of root-knot nematodes in greenhouse confirmed previous data by Greco et al. [5]. Moreover, the yield increases obtained in our greenhouse (65.5% after the first crop and 50.5–62.1% on the second crop), that had low to medium infestation level, are similar to the increase averages reported by Greco et al. [5] of 42.3% in low infested soil and 79.2% in medium soil infestation level.

Information on the long-lasting effects of one fumigation treatment in more successive crops is also useful for economical and practical agronomic decisions. Our study clearly demonstrates the effectivity of DMDS alone and in strategies of over two-three crop cycles per year, confirming the previous results of Toth et al. [20].

In addition to nematicide activity, DMDS has also fungicide and herbicide activities. The fungicide efficiency of DMDS was demonstrated against various soil-borne pathogens e.g., *Athelia rolfsii* (Curzi), *Colletotrichum coccodes* (Wallr.), *Fusarium oxysporum* Schlectend., *Macrophomina phaseolina* (Tassi), *Rhizoctonia solani* Kühn, *Pythium ultimum* Trow., *Phytophthora cactorum* (Lebert et Cohn), *Sclerotinia sclerotiorum* (Lib.) and *Verticillium dahliae* Kleb. [21–30]. The herbicide activity of DMDS has been demonstrated by several authors [21,24,29,31].

In the Bulgarian trial (2018–2019), the study on the soil dynamics of the nematode, from 3 July 2018 until 8 October 2019, in a greenhouse cropped to several cycles, has confirmed the long-lasting effect of DMDS in controlling root-knot nematode over five consecutive crops (cucumber, lettuce, lettuce, cucumber, cucumber), but three cucumber crops as an important host for RKN. In the studied greenhouse trials, the soil nematode population in the plots fumigated with DMDS remained low until the end of the third crop cycles (sampling 5), exactly until one year after the treatments, and increased slightly only thereafter. Instead, RKN population remained at rather high level throughout different crop cycles both in the control or non-fumigated plots.

Although a number of studies on the effect of nematicide on non-target nematodes were conducted in open field on different crops [32–35], very few investigations were conducted in a greenhouse. Specific conditions in greenhouse production, e.g., higher temperatures and excessive N fertilization, decrease of pH, monoculture or crop rotations with known susceptible host plant species can result in the rapid accumulation of RKNs and loss of diversity and richness [36–38]. Our data on nematode trophic web conditions confirm that low diversity and less evenness were observed in the entire greenhouse area. The nematode food web revealed a negative effect 18 days after application of DMDS and a recovery to its initial status by three months after the end of the first crop. The prevalence of bacterial decomposition channels and low numbers of fungal feeding trophic groups were also observed by Liu et al. [36].

In conclusion, DMDS can be considered to possess great efficacy in controlling root-knot nematodes in greenhouses in Bulgaria. It is noteworthy to mention that the control strategy with DMDS before the first crop was followed by biological means of control (garlic extract and *Trichoderma* spp.) on the succeeding crops, and nearly led to a crop that was free of plant-parasitic nematodes. These results were beneficial for the following second crop and still had significant control effect on the root-knot nematode population over the third crop. On the other hand, the negative impact of the fumigation with DMDS on non-target nematodes is very limited in time as nematode communities regained the previous crop level in a matter of a few months (five in our study). Therefore, DMDS can be properly included in soil IPM programmes and appears very promising for the protected crop industry in Europe.

**Author Contributions:** Conceptualization, Z.I., A.M. (Arben Myrta); evaluation of RGI and yield A.M. (Alexander Mitev); evaluation soil stages of RKN and free-living nematodes, T.L., Z.I.; data curation and analysis T.L. and Z.I.; writing—original draft preparation Z.I.; writing—review and editing, A.M. (Arben Myrta). All authors have read and agreed to the published version of the manuscript.

**Funding:** Study was supported at 80% by ISSAPP and Certis EUROPE according their agreement from 1 June 2018. External funding on analysis of soil nematode community 20% was received by Bulgarian Science Fund, grant No КП-06 H36/12 2019, project with acronym GallNem.

**Institutional Review Board Statement:** Not applicable.

**Informed Consent Statement:** Not applicable.

**Data Availability Statement:** Data are available on request.

**Acknowledgments:** We would like to thank to Georgi Gergov from KNE Certis Bulgaria for logistics connected with the greenhouse owner and DMDS application, and Nicola Greco, formerly at CNR, Institute of Sustainable Plant Protection, Bari, Italy, for his critical reading and suggestions during the preparation of this paper.

**Conflicts of Interest:** The authors declare no conflict of interest.

**Appendix A**

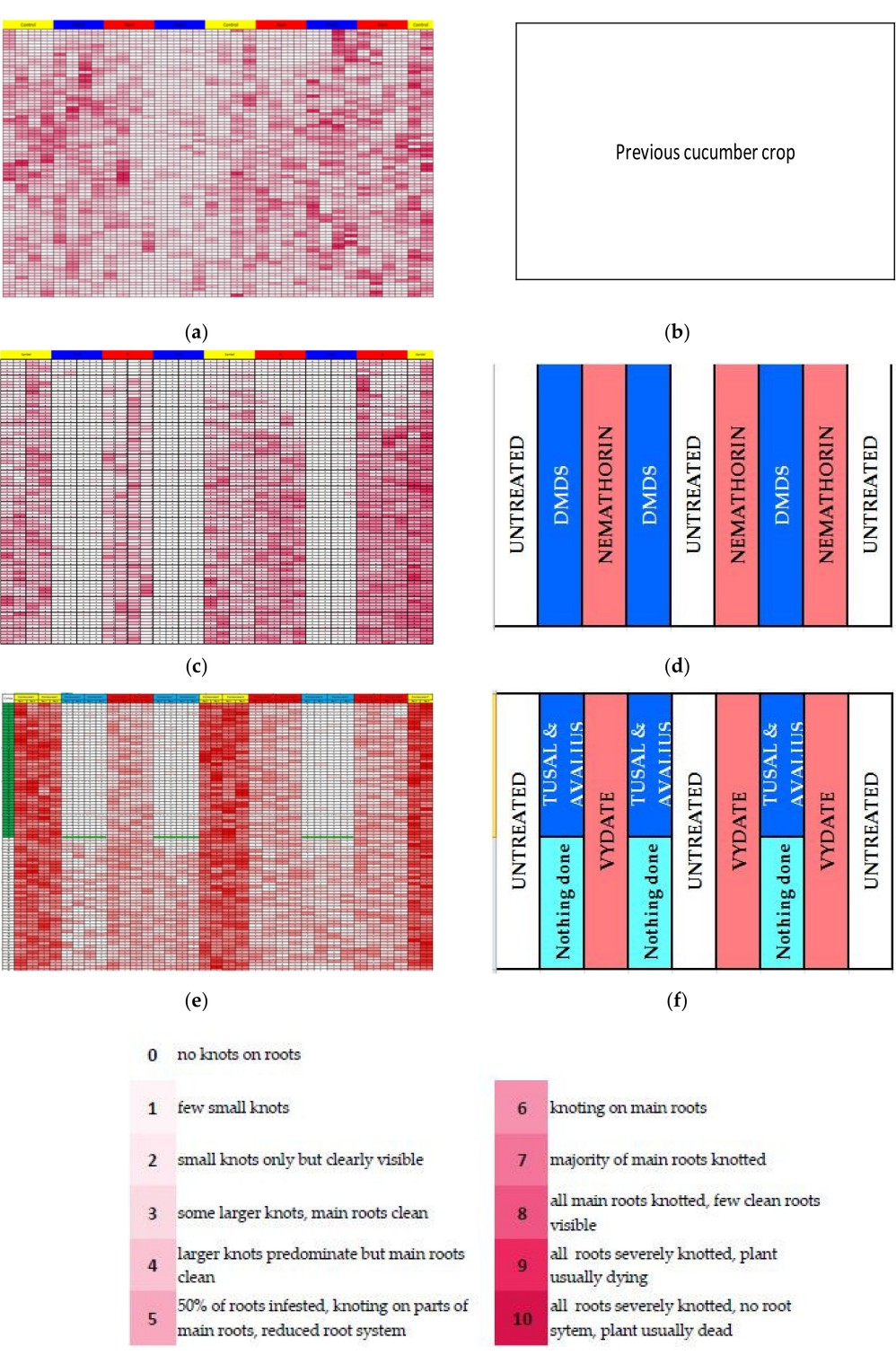

**Figure A1.** Map of *Meloidogyne* spp. root-knot infestation level (estimated according to Bridge and Page [14] rating scale of root-galling) and schemes of respective experimental plots. (**a**) Root mapping before starting the trial (previous crop), (**b**) Plot selected for the trial. (**c**) Root mapping at the end of first crop. (**d**) Scheme of the trial at first cucumber crop. (**e**) Root mapping at the end of second crop. (**f**) Scheme of the trial at second cucumber crop.

# Appendix B

**Table A1.** Structure, abundance and occurrence of plant-parasitic nematodes in trial variants during the study period.

| Guild | Genus | 1 DBA Mean | 1 DBA Occurence | 18 DAA Mean | 18 DAA Occurence | 104 DAA Mean | 104 DAA Occurence | 258 DAA Mean | 258 DAA Occurence | 362 DAA Mean | 362 DAA Occurence | 450 DAA Mean | 450 DAA Occurence |
|---|---|---|---|---|---|---|---|---|---|---|---|---|---|
| **Untreated** | | | | | | | | | | | | | |
| Pp-a-3 | *Meloidogyne* | 401.8 | 100% | 246.0 | 100% | 454.1 | 100% | 94.3 | 100% | 93.9 | 100% | 268.8 | 100% |
| Pp-d-2 | *Paratylenchus* | 4.7 | 33% | 3.7 | 58% | | | | | 1.2 | 42% | 2.4 | 42% |
| Pp-d-3 | *Criconemella* | 4.7 | 17% | 1.2 | 25% | | | | | 0.9 | 8% | 2.4 | 17% |
| Pp-d-3 | *Tylenchorhynchus* | | | 0.2 | 8% | | | | | | | | |
| Pp-e-2 | *Psilenchus* | 4.7 | 33% | 5.5 | 75% | | | | | | | 2.3 | 42% |
| Total | | 415.9 | | 256.7 | | 454.1 | | 94.3 | | 96.1 | | 275.9 | |
| **Non-fumigant Nematicide** | | | | | | | | | | | | | |
| Pp-a-3 | *Meloidogyne* | 358.8 | 100% | 153.4 | 100% | 640.7 | 100% | 367.0 | 100% | 138.1 | 92% | 295.0 | 100% |
| Pp-d-2 | *Paratylenchus* | 6.6 | 42% | 1.8 | 42% | | | | | 0.9 | 25% | 3.1 | 67% |
| Pp-d-3 | *Criconemella* | 6.6 | 33% | 2.7 | 33% | 10.5 | 8% | 2.1 | 17% | 3.4 | 8% | 6.1 | 25% |
| Pp-d-3 | *Tylenchorhynchus* | | | 0.5 | 8% | | | | | | | | |
| Pp-e-2 | *Psilenchus* | 1.9 | 17% | 3.0 | 33% | | | | | 0.2 | 8% | 0.8 | 17% |
| Total | | 373.9 | | 161.3 | | 651.2 | | 369.1 | | 142.6 | | 305.0 | |
| **DMDS** | | | | | | | | | | | | | |
| Pp-a-3 | *Meloidogyne* | 328.2 | 100% | 10.5 | 75% | 14.0 | 50% | 43.8 | 58% | 6.0 | 50% | 113.1 | 92% |
| Pp-d-2 | *Paratylenchus* | 8.5 | 25% | | | | | | | 0.7 | 17% | 3.6 | 67% |
| Pp-d-3 | *Criconemella* | 0.9 | 8% | | | | | | | | | 0.2 | 8% |
| Pp-d-3 | *Tylenchorhynchus* | 0.9 | 8% | | | | | 4.5 | 58% | 0.2 | 8% | 0.3 | 8% |
| Pp-e-2 | *Psilenchus* | 1.9 | 17% | | | | | | | | | 0.2 | 8% |
| Total | | 340.4 | | 10.5 | | 14.0 | | 48.3 | | 6.9 | | 117.3 | |

**Table A2.** Bacterial feeding nematodes structure. Abundance and occurrence in trial variants during the study period.

| Guild | Genus | 1 DBA Mean | 1 DBA Occurence | 18 DAA Mean | 18 DAA Occurence | 104 DAA Mean | 104 DAA Occurence | 258 DAA Mean | 258 DAA Occurence | 362 DAA Mean | 362 DAA Occurence | 450 DAA Mean | 450 DAA Occurence |
|---|---|---|---|---|---|---|---|---|---|---|---|---|---|
| **Untreated** | | | | | | | | | | | | | |
| Ba 1 | *Cruznema* | 69.4 | 83% | 36.3 | 100% | 435.4 | 67% | 10.2 | 67% | 13.2 | 75% | 104.8 | 92% |
| Ba 1 | *Cuticularia* | | | | | | | 0.5 | 17% | 3.6 | 33% | | |
| Ba 1 | *Diplagaster* | | | | | | | | | | | | |
| Ba 1 | *Diploscapter* | 68.9 | 83% | 39.0 | 92% | | | | | 48.3 | 92% | 54.7 | 92% |
| Ba 1 | *Mesorhabditis* | 51.2 | 67% | 18.7 | 75% | | | 0.7 | 25% | 0.5 | 8% | 33.1 | 67% |
| Ba 1 | *Panagrolaimus* | 6.6 | 33% | 16.9 | 67% | 117.7 | 25% | 43.6 | 92% | 1.9 | 17% | 6.3 | 42% |
| Ba 1 | *Rhabditidae* | | | 96.5 | 100% | | | | | 187.8 | 100% | | |
| Ba 2 | *Acrobeles* | 16.1 | 75% | 12.4 | 83% | | | 5.7 | 58% | 4.1 | 58% | 10.9 | 83% |
| Ba 2 | *Acrobeloides* | 44.7 | 83% | 16.1 | 100% | | | 6.1 | 83% | 1.7 | 42% | 32.4 | 83% |
| Ba 2 | *Cephalobus* | 6.6 | 42% | 5.1 | 58% | | | 1.0 | 8% | 4.1 | 25% | 10.9 | 67% |
| Ba 2 | *Chiloplacus* | 1.9 | 25% | 3.9 | 42% | | | 16.8 | 100% | 5.9 | 42% | 3.8 | 25% |
| Ba 2 | *Heterocephalobus* | 7.5 | 50% | 8.3 | 83% | | | | | 13.6 | 75% | 10.4 | 75% |
| Ba 2 | *Plectus* | 3.8 | 33% | 8.2 | 58% | | | | | 1.1 | 17% | 2.7 | 50% |
| Ba 2 | *Ypsilonellus* | 0.9 | 8% | 3.3 | 42% | | | | | | | 0.4 | 8% |
| Ba 2 | *Zeldia* | 156.5 | 100% | 93.6 | 100% | 49.8 | 92% | 118.0 | 100% | 76.1 | 100% | 171.1 | 100% |
| Ba 3 | *Acromadora* | | | 0.2 | 8% | | | | | 0.5 | 17% | | |
| Ba 3 | *Prismatolaimus* | 8.5 | 50% | 5.8 | 67% | 3.1 | 42% | 1.2 | 42% | 2.3 | 58% | 5.2 | 75% |
| Ba 3 | *Teratocephalus* | | | 0.2 | 8% | 1.1 | 8% | | | | | | |
| Ba 4 | *Alaimus* | | | | | | | | | | | | |
| Total | | 442.7 | | 364.7 | | 607.3 | | 203.8 | | 364.5 | | 446.8 | |

**Table A2.** *Cont.*

| | | 1 DBA | | 18 DAA | | 104 DAA | | 258 DAA | | 362 DAA | | 450 DAA | |
|---|---|---|---|---|---|---|---|---|---|---|---|---|---|
| Guild | Genus | Mean | Occurrence | Mean | Occurrence | Mean | Occurrence | Mean | Occurrence | Mean | Occurrence | Mean | Occurrence |
| **Non-fumigant Nematicide** | | | | | | | | | | | | | |
| Ba 1 | *Cruznema* | 78.8 | 100% | 36.9 | 100% | | | 16.1 | 67% | 46.6 | 92% | 84.4 | 100% |
| Ba 1 | *Cuticularia* | | | | | | | 0.3 | 17% | 4.7 | 50% | | |
| Ba 1 | *Diplagaster* | | | 0.2 | 8% | | | | | 0.9 | 17% | | |
| Ba 1 | *Diploscapter* | 36.9 | 83% | 24.1 | 100% | | | | | 18.8 | 50% | 11.0 | 83% |
| Ba 1 | *Mesorhabditis* | 16.0 | 42% | 16.1 | 75% | | | 1.5 | 42% | 6.0 | 58% | 6.1 | 42% |
| Ba 1 | *Panagrolaimus* | 13.1 | 25% | 13.3 | 50% | 545.5 | 100% | 25.2 | 100% | 1.3 | 25% | 1.4 | 17% |
| Ba 1 | *Rhabditidae* | | | 123.9 | 100% | | | | | 160.9 | 100% | | |
| Ba 2 | *Acrobeles* | 14.2 | 58% | 12.2 | 75% | | | 4.2 | 75% | 9.8 | 83% | 9.4 | 67% |
| Ba 2 | *Acrobeloides* | 16.1 | 75% | 9.5 | 67% | | | 3.5 | 67% | 6.7 | 67% | 18.5 | 92% |
| Ba 2 | *Cephalobus* | 5.7 | 50% | 3.9 | 67% | | | | | 0.7 | 25% | 13.0 | 92% |
| Ba 2 | *Chiloplacus* | | | 7.7 | 67% | | | 32.5 | 100% | 2.8 | 67% | 0.4 | 8% |
| Ba 2 | *Heterocephalobus* | 2.8 | 25% | 5.0 | 67% | | | 0.2 | 8% | 14.4 | 92% | 4.4 | 83% |
| Ba 2 | *Plectus* | 7.6 | 42% | 19.6 | 83% | | | 0.2 | 8% | | | 5.5 | 67% |
| Ba 2 | *Ypsilonellus* | | | 3.4 | 50% | | | | | | | | |
| Ba 2 | *Zeldia* | 111.0 | 100% | 89.8 | 100% | 78.7 | 83% | 273.5 | 100% | 65.9 | 100% | 130.5 | 100% |
| Ba 3 | *Acromadora* | | | | | | | | | 0.2 | | | |
| Ba 3 | *Prismatolaimus* | 4.7 | 42% | 4.1 | 58% | 1.0 | 8% | | | 0.5 | 17% | 1.9 | 67% |
| Ba 3 | *Teratocephalus* | | | | | 0.9 | 8% | 0.1 | 8% | | | | |
| Ba 4 | *Alaimus* | | | | | | | | | 0.2 | | | |
| Total | | 306.9 | | 369.7 | | 626.1 | | 357.4 | | 340.4 | | 286.6 | |
| **DMDS** | | | | | | | | | | | | | |
| Ba 1 | *Cruznema* | 58.9 | 92% | | | | | 3.6 | 58% | 3.8 | 42% | 45.2 | 100% |
| Ba 1 | *Cuticularia* | | | | | | | 3.9 | 50% | 0.7 | 8% | | |
| Ba 1 | *Diplagaster* | | | | | | | | | | | | |
| Ba 1 | *Diploscapter* | 28.2 | 83% | 0.5 | 17% | | | | | 39.6 | 83% | 22.1 | 92% |
| Ba 1 | *Mesorhabditis* | 36.5 | 50% | | | | | 2.9 | 58% | 0.2 | 8% | 39.6 | 50% |
| Ba 1 | *Panagrolaimus* | 0.9 | 8% | 0.7 | 25% | 64.6 | 100% | 127.0 | 100% | | | 1.8 | 17% |
| Ba 1 | *Rhabditidae* | | | | | 13.4 | 50% | | | 48.9 | 100% | | |
| Ba 2 | *Acrobeles* | 29.1 | 33% | | | | | 5.7 | 67% | 6.8 | 50% | 26.4 | 58% |
| Ba 2 | *Acrobeloides* | 26.4 | 67% | 0.5 | 17% | | | 62.7 | 100% | 8.8 | 83% | 16.0 | 100% |
| Ba 2 | *Cephalobus* | 4.7 | 25% | | | | | 0.9 | 17% | 5.7 | 8% | 4.7 | 50% |
| Ba 2 | *Chiloplacus* | 1.9 | 17% | | | | | 6.0 | 92% | 3.3 | 50% | 5.4 | 42% |
| Ba 2 | *Heterocephalobus* | 2.8 | 25% | | | | | 0.7 | 8% | 0.5 | 17% | 2.1 | 50% |
| Ba 2 | *Plectus* | 0.9 | 8% | | | | | 0.2 | 8% | 2.5 | 50% | 0.8 | 25% |
| Ba 2 | *Ypsilonellus* | 1.9 | 8% | | | | | | | | | | |
| Ba 2 | *Zeldia* | 178.9 | 100% | 6.2 | 75% | 115.0 | 100% | 54.2 | 100% | 70.5 | 100% | 237.3 | 100% |
| Ba 3 | *Acromadora* | | | | | | | | | | | | |
| Ba 3 | *Prismatolaimus* | 13.2 | 75% | | | 0.5 | 8% | 0.6 | 0% | 0.7 | 25% | 3.0 | 67% |
| Ba 3 | *Teratocephalus* | | | | | | | 0.5 | 25% | | | | |
| Ba 4 | *Alaimus* | | | | | | | | | | | | |
| Total | | 384.2 | | 7.8 | | 193.5 | | 269.0 | | 192.0 | | 404.2 | |

**Table A3.** Fungal feeding nematodes structure. Abundance and occurrence in trial variants during the study period.

| | | 1 DBA | | 18 DAA | | 104 DAA | | 258 DAA | | 362 DAA | | 450 DAA | |
|---|---|---|---|---|---|---|---|---|---|---|---|---|---|
| Guild | Genus | Mean | Occurrence | Mean | Occurrence | Mean | Occurrence | Mean | Occurrence | Mean | Occurrence | Mean | Occurrence |
| **Untreated** | | | | | | | | | | | | | |
| Fu 2 | *Aphelenchoides* | 9.5 | 33% | 2.8 | 67% | | | 0.9 | 25% | | | 5.7 | 33% |
| Fu 2 | *Aphelenchus* | 7.5 | 42% | 23.0 | 67% | 4.1 | 25% | 4.0 | 67% | 12.9 | 100% | 3.8 | 42% |
| Fu 2 | *Ditylenchus* | | | 1.8 | 33% | | | | | | | | |
| Fu 2 | *Filenchus* | | | | | 1.1 | 8% | | | | | | |
| Fu 2 | *Lelenchus* | | | | | | | | | 0.8 | 25% | | |
| Fu 2 | *Paraphelenchus* | 11.3 | 58% | 6.2 | 75% | | | | | | | 10.7 | 67% |
| Total | | 28.3 | | 33.8 | | 5.3 | | 5.0 | | 13.7 | | 20.2 | |

**Table A3.** *Cont.*

| | | Sampling | | | | | | | | | | | |
|---|---|---|---|---|---|---|---|---|---|---|---|---|---|
| | | 1 DBA | | 18 DAA | | 104 DAA | | 258 DAA | | 362 DAA | | 450 DAA | |
| Guild | Genus | Mean | Occurrence | Mean | Occurrence | Mean | Occurrence | Mean | Occurrence | Mean | Occurrence | Mean | Occurrence |
| **Non-fumigant Nematicide** | | | | | | | | | | | | | |
| Fu 2 | *Aphelenchoides* | 8.4 | 17% | 5.2 | 42% | | | 1.6 | 25% | | | 2.8 | 17% |
| Fu 2 | *Aphelenchus* | 14.1 | 75% | 28.5 | 100% | 1.9 | 8% | 6.0 | 75% | 24.1 | 67% | 5.2 | 75% |
| Fu 2 | *Ditylenchus* | 1.0 | 8% | 0.5 | 17% | | | | | | | 0.2 | 8% |
| Fu 2 | *Filenchus* | 1.9 | 8% | | | 1.9 | 17% | 0.1 | 8% | | | | |
| Fu 2 | *Lelenchus* | | | | | | | | | | | | |
| Fu 2 | *Paraphelenchus* | | | 6.1 | 25% | | | | | | | | |
| Total | | 25.4 | | 40.3 | | 3.9 | | 7.7 | | 24.1 | | 8.2 | |
| **DMDS** | | | | | | | | | | | | | |
| Fu 2 | *Aphelenchoides* | | | | | 0.7 | 17% | | | | | | |
| Fu 2 | *Aphelenchus* | 3.8 | 25% | | | 1.4 | 8% | 0.9 | 42% | 0.7 | 17% | 0.6 | 17% |
| Fu 2 | *Ditylenchus* | | | | | | | | | | | | |
| Fu 2 | *Filenchus* | | | | | 0.2 | 8% | 0.1 | 8% | 0.3 | 8% | | |
| Fu 2 | *Lelenchus* | | | | | | | 0.7 | 17% | | | | |
| Fu 2 | *Paraphelenchus* | 0.9 | 8% | | | | | | | | | 0.6 | 17% |
| Total | | 4.7 | | 0.0 | | 2.4 | | 1.7 | | 0.9 | | 1.2 | |

**Table A4.** Omnivorous nematodes—structure. Abundance and occurrence in trial variants during the study period.

| | | Sampling | | | | | | | | | | | |
|---|---|---|---|---|---|---|---|---|---|---|---|---|---|
| | | 1 DBA | | 18 DAA | | 104 DAA | | 258 DAA | | 362 DAA | | 450 DAA | |
| Guild | Genus | Mean | Occurrence | Mean | Occurrence | Mean | Occurrence | Mean | Occurrence | Mean | Occurrence | Mean | Occurrence |
| **Untreated** | | | | | | | | | | | | | |
| Om4 | *Ecumenicus* | 2.8 | 25% | 0.9 | 33% | | | | | 0.2 | 8% | 1.2 | 25% |
| Om4 | *Eudorylaimus* | | | 0.5 | 17% | | | | | | | | |
| Om4 | *Mesodorylaimus* | 1.9 | 17% | 0.9 | 33% | | | | | 2.8 | 42% | 0.8 | 17% |
| Om4 | *Thonus* | 7.6 | 58% | 4.1 | 83% | | | 0.7 | 17% | | | 4.0 | 58% |
| Om5 | *Aporcelaimus* | 0.9 | 8% | 0.2 | 8% | | | | | | | 0.5 | 8% |
| Om5 | *Metaporcelaimus* | 2.9 | 25% | 1.8 | 42% | | | | | | | 2.6 | 42% |
| Om5 | *Aporcelaimellus* | 3.8 | 33% | 2.1 | 58% | | | 4.1 | 58% | 0.4 | 17% | 3.0 | 42% |
| Om5 | *Aporcella* | | | | | 2.4 | 8% | 0.2 | 8% | | | | |
| Total | | 19.9 | | 10.5 | | 2.4 | | 5.0 | | 3.4 | | 12.2 | |
| **Non-fumigant Nematicide** | | | | | | | | | | | | | |
| Om4 | *Ecumenicus* | 4.7 | 33% | 1.4 | 33% | | | 10.8 | 17% | 0.4 | 17% | 1.2 | 42% |
| Om4 | *Eudorylaimus* | | | 0.5 | 17% | | | 0.1 | 8% | | | | |
| Om4 | *Mesodorylaimus* | 2.9 | 25% | 2.8 | 42% | | | 0.8 | 25% | 5.3 | 50% | 2.4 | 33% |
| Om4 | *Thonus* | 8.5 | 50% | 3.7 | 67% | 1.9 | 17% | 1.7 | 42% | | | 1.8 | 58% |
| Om5 | *Aporcelaimus* | | | 0.2 | 8% | 2.4 | 25% | | | | | | |
| Om5 | *Metaporcelaimus* | | | | | | | | | | | | |
| Om5 | *Aporcelaimellus* | 7.6 | 50% | 2.2 | 42% | | | 1.1 | 42% | | | 2.2 | 50% |
| Om5 | *Aporcella* | 3.7 | 17% | 1.1 | 33% | | | | | 0.2 | 8% | 0.3 | 8% |
| Total | | 27.4 | | 11.9 | | 4.3 | | 14.6 | | 5.9 | | 8.0 | |
| **DMDS** | | | | | | | | | | | | | |
| Om4 | *Ecumenicus* | 2.8 | 25% | | | | | 0.5 | 17% | 0.7 | 25% | 0.7 | 25% |
| Om4 | *Eudorylaimus* | 0.9 | 8% | | | | | 0.2 | 17% | | | 0.2 | 8% |
| Om4 | *Mesodorylaimus* | 1.9 | 8% | | | 0.5 | 17% | 0.1 | 8% | 0.2 | 8% | | |
| Om4 | *Thonus* | 4.7 | 42% | | | | | 0.8 | 17% | 0.2 | 8% | 4.0 | 42% |
| Om5 | *Aporcelaimus* | 0.9 | 8% | | | | | | | | | | |
| Om5 | *Metaporcelaimus* | 6.6 | 42% | | | | | | | 0.7 | 8% | 3.0 | 58% |
| Om5 | *Aporcelaimellus* | 11.3 | 50% | | | | | 1.0 | 42% | 0.5 | 17% | 5.8 | 75% |
| Om5 | *Aporcella* | 5.6 | 42% | | | 0.5 | 8% | | | | | 1.6 | 42% |
| Total | | 34.8 | | 0.0 | | 1.0 | | 2.7 | | 2.3 | | 15.4 | |

**Table A5.** Animal predator nematodes structure. Abundance and occurrence in trial variants during study period.

| | | Sampling | | | | | | | | | | | |
|---|---|---|---|---|---|---|---|---|---|---|---|---|---|
| | | 1 DBA | | 18 DAA | | 104 DAA | | 258 DAA | | 362 DAA | | 450 DAA | |
| Guild | Genus | Mean | Occurrence | Mean | Occurrence | Mean | Occurrence | Mean | Occurrence | Mean | Occurrence | Mean | Occurrence |
| **Untreated** | | | | | | | | | | | | | |
| Ca-2 | *Seinura* | 1.9 | 8% | 0.7 | 8% | | | | | | | 0.8 | 17% |
| Ca-3 | *Tobrilus* | 0.9 | 8% | 0.2 | 8% | | | | | 0.2 | 8% | | |
| Ca-4 | *Clarcus* | | | | | | | | | | | | |
| Ca-4 | *Mononchus* | 1.9 | 17% | 0.7 | 17% | | | | | | | 1.3 | 17% |
| Ca-4 | *Mylonchulus* | 3.8 | 25% | 3.0 | 58% | | | | | 0.2 | 8% | 2.5 | 33% |
| Ca-5 | *Carcharolaimus* | | | | | | | | | | | | |
| Ca-5 | *Sectonema* | | | | | | | | | | | | |
| Total | | 8.5 | | 4.6 | | 0.0 | | 0.0 | | 0.3 | | 4.6 | |
| **Non-fumigant Nematicide** | | | | | | | | | | | | | |
| Ca-2 | *Seinura* | | | 0.4 | 8% | | | | | 1.5 | 8% | | |
| Ca-3 | *Tobrilus* | 1.0 | 8% | 0.7 | 17% | | | | | | | 0.2 | 8% |
| Ca-4 | *Clarcus* | | | 0.2 | 8% | | | | | | | | |
| Ca-4 | *Mononchus* | 0.9 | 8% | 0.4 | 8% | | | | | | | 0.2 | 8% |
| Ca-4 | *Mylonchulus* | 1.9 | 17% | 0.7 | 25% | | | | | 0.9 | 33% | 0.4 | 17% |
| Ca-5 | *Carcharolaimus* | | | | | | | | | | | 0.7 | 17% |
| Ca-5 | *Sectonema* | 0.9 | 8% | 0.4 | 8% | | | | | | | 0.2 | 8% |
| Total | | 4.7 | | 3.0 | | 0.0 | | 0.0 | | 2.4 | | 1.7 | |
| **DMDS** | | | | | | | | | | | | | |
| Ca-2 | *Seinura* | | | | | | | | | | | | |
| Ca-3 | *Tobrilus* | | | | | | | | | | | | |
| Ca-4 | *Clarcus* | | | | | | | | | 0.2 | 8% | | |
| Ca-4 | *Mononchus* | | | | | 0.2 | 8% | | | | | | |
| Ca-4 | *Mylonchulus* | 1.0 | 8% | | | | | | | | | 0.2 | 8% |
| Ca-5 | *Carcharolaimus* | | | | | | | | | | | | |
| Ca-5 | *Sectonema* | | | | | | | | | | | | |
| Total | | 1.0 | | 0.0 | | 0.2 | | 0.0 | | 0.2 | | 0.2 | |

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
