# Peer review of "Monitoring of Long-Lasting Effects of Fumigation with Dimethyl Disulfide (DMDS) on Root-Gall Index, Root-Knots, Other Nematode Populations, and Crop Yield over Three Protected Cucumber Crops in Bulgaria"

_agronomy, doi:10.3390/agronomy11061206_

Round 1
Reviewer 1 Report
See attached file.

Author Response
Dear Reviewer 1,
Thank you for review. Your recomendations help us to improve manuscript clarity. You can find some explanations to your questions and recomendations.
R1: Section 3.2 states ‘Root gall index on the previous cucumber crop …… was rather uniform ‘ – but in the figure (which now will be A1a) this is clearly not the case. There’s a ‘low’ vertical stripe down the mid-left, and notable scattered local hot-spots. These ‘pre-existing’ values should be included in all analyses as a possible covariate. The stats methods section is quite brief, but does not mention any testing of covariates. If the covariate is significant (or near-so), it will improve the analyses. If the covariate is not significant, state this ‘to demonstrate you have conducted full investigations’.
When we stated rather uniform we meant usual more patched distribution of RKN. In studied glasshouse over 7 year cucumber was cropped continuously (e.g. Site description) that resulted in more uniform distribution. Locality as covariate of date in different variants was tested now but not significant (P=0.0673).
- Where is Appendix A? Appendix B (the figures) needs to be changed to Appendix A; and then Appendix C (the tables) to Appendix B. Curiously, when talking about these tables (Section 3.5), the text correctly calls them ‘Appendix B’.
Appendices have been numbered as recommended.
- The use of * in Figure 1 and 2 is ambiguous – you need to specify which of the three means (at each date) are different from which. Using letters (as in the tables) is the accepted way of showing this.
Significant differences in Figure 1 and 2 have been marked with letters as recommended.
In addition, there are some editorial-style improvements to be made –
All editorial improvements were implemented in the text
Reviewer 2 Report
The Manuscript regards the use of DMDS in cucumber and lettuce protected conditions against root-knot nemtodes (RKNs). In the trials carried out in the 2018-2019 period the efficacy of the fumigant DMDS on RKNs was evaluated also in a long term period to protect following crops. Moreover in this study other nematodes groups (bacterial and fungal feeders nematodes, omnivores and animal predators nematodes) were considered as biondicators of the healthy soil conditions following the use of DMDS.
The manuscript is well organised and exhaustive of many aspect with reference to nematode control, identification, sampling, mapping of the greenhouse soil for its nematode infestation level, use of nematodes as bioindicators and so on. The Tables are well organised and clear as well as the maps relative to the nematode infestation levels during the trials period and the figures.
However, in the text there were some inaccuracies. All these inaccuracies are well evidenciated in yellow or green colors in the uploaded pdf file of the manuscript.
Moreover, the Authors wrote about Appendix B and C jumping A. So, as reported in the manuscript I suggest them to insert Appendix A instead B and Appendix B instead C.

Author Response
Dear Reviewer 2,
Thank you for thorough review of our manuscript that helps to improve text distinctness and clarity.
There are some explanations that we should give on your questions:
67-68 Trials were conducted in glasshouse. We replace greenhouse term from that paragraph.
89 - DMDS fumigant was applied before first cucumber transplanting.
156 - Rhabditidae is a name of family rang and italic letters are applied only for genus and species level according International Zoological Code (see Notton et al., 2011; https://www.iczn.org/)